# A Positive Regulatory Feedback Loop between EKLF/KLF1 and TAL1/SCL Sustaining the Erythropoiesis

**DOI:** 10.3390/ijms22158024

**Published:** 2021-07-27

**Authors:** Chun-Hao Hung, Tung-Liang Lee, Anna Yu-Szu Huang, Kang-Chung Yang, Yu-Chiau Shyu, Shau-Ching Wen, Mu-Jie Lu, Shinsheng Yuan, Che-Kun James Shen

**Affiliations:** 1Institute of Molecular Biology, Academia Sinica, Taipei 11529, Taiwan; libur777@gate.sinica.edu.tw (C.-H.H.); andywork0711@gmail.com (T.-L.L.); annabellehuang7645@gmail.com (A.Y.-S.H.); yuchiaushyu@gmail.com (Y.-C.S.); sophiweng62@gmail.com (S.-C.W.); ruru79320@gmail.com (M.-J.L.); 2Department of Life Science, Institute of Genomic Research, National Yang-Ming University, Taipei 112, Taiwan; 3Institute of Molecular Medicine, College of Medicine, National Taiwan University, Taipei 10617, Taiwan; 4Genome and Systems Biology Degree Program, College of Life Science, National Taiwan University, Taipei 10617, Taiwan; kcyang@stat.sinica.edu.tw; 5Institute of Statistical Science, Academia Sinica, Taipei 11529, Taiwan; 6Community Medicine Research Center, Chang Gang Memorial Hospital, Keelung Branch, Keelung 204, Taiwan; 7Department of Nursing, Chang Gang University of Science and Technology, Taoyuan City 333, Taiwan; 8The PhD Program for Neural Regenerative Medicine, Taipei Medical University, Taipei 110, Taiwan

**Keywords:** erythroid differentiation, EKLF/KLF1, gene knockout, TAL1/SCL, global gene expression profiling, direct target genes, genomic footprinting of *Tal*1 promoter, transcriptional regulation

## Abstract

The erythroid Krüppel-like factor EKLF/KLF1 is a hematopoietic transcription factor binding to the CACCC DNA motif and participating in the regulation of erythroid differentiation. With combined use of microarray-based gene expression profiling and the promoter-based ChIP-chip assay of E14.5 fetal liver cells from wild type (WT) and EKLF-knockout (*Eklf*^−/−^) mouse embryos, we identified the pathways and direct target genes activated or repressed by EKLF. This genome-wide study together with the molecular/cellular analysis of the mouse erythroleukemic cells (MEL) indicate that among the downstream direct target genes of EKLF is *Tal1*/*Scl*. *Tal1*/*Scl* encodes another DNA-binding hematopoietic transcription factor TAL1/SCL, known to be an *Eklf* activator and essential for definitive erythroid differentiation. Further identification of the authentic Tal gene promoter in combination with the in vivo genomic footprinting approach and DNA reporter assay demonstrate that EKLF activates the Tal gene through binding to a specific CACCC motif located in its promoter. These data establish the existence of a previously unknow positive regulatory feedback loop between two DNA-binding hematopoietic transcription factors, which sustains mammalian erythropoiesis.

## 1. Introduction

Erythropoiesis is a dynamic process sustained throughout the whole lifetime of vertebrates for the generation of red blood cells from pluripotent hematopoietic stem cells (HSCs). In the ontogeny of mouse erythropoiesis, the major locations of HSC change orderly several times, convert from embryonic yolk sac to fetal liver, and then to the spleen and bone marrow in adult mice [1]. At each of these tissues, the multistep differentiation process of erythropoiesis is accompanied with a series of lineage-specific activations and the restriction of gene expression, as mediated by several erythroid-specific/erythroid-enriched transcription factors, including GATA1, TAL1/SCL, NF-E2, and EKLF [2,3,4,5]. Among the factors regulating erythropoiesis is the Erythroid Krüppel-like factor (EKLF/KLF1), a pivotal regulator that functions in erythroid differentiation and fate decision through the bipotential megakaryocyte-erythroid progenitors (MEPs) [6,7,8,9,10], as well as the homeostasis of HSC [11]. *Eklf* is the first identified member of the KLF family of genes expressed in the erythroid cells, mast cells, and their precursors [12,13], as well as some of the other types of the hematopoietic cells, but at low levels [6,14] (Bio GPS). The critical function of *Eklf* in erythropoiesis was initially demonstrated through gene abolition studies, with the *Eklf*-knockout mice (*Eklf*^−/−^) displaying severe anemia and dying in utero at around embryonic (E) day 14.5 (E14.5) [15,16]. The impairment of the definitive erythropoietic differentiation is a major cause of embryonic lethality in *Eklf*^−/−^ mice, in addition to β-thalassemia [17,18].

EKLF regulates its downstream genes, including the adult β globin genes, through binding of its C-terminal C_2_H_2_ zinc finger domain to the canonical binding sequence CCNCNCCC located in the promoters or enhancers [18,19,20] and the recruitment of co-activators, e.g., CBP/p300 [21] and SWI/SNF-related chromatin remodeling complex [22,23], or co-repressors [7,18]. Moreover, clinical associations exist between the *Eklf* gene and different human hematopoietic phenotypes or diseases [10,24,25]. In erythroid progenitors, e.g., CFU-E and Pro-E, EKLF is mainly located in the cytoplasm. Upon differentiation of Pro-E to Baso-E, EKLF is imported into the nucleus and forms distinct nuclear bodies [20,26]. Genome-wide analysis of the global functions of mouse EKLF through the identification of the direct transcription target genes has been conducted using ChIP-Seq in combination with gene expression profiling [27,28]. The results from these studies suggest that EKLF functions mainly as a transcription activator in cooperation with TAL1/SCL and/or GATA1 to target genes including those required for terminal erythroid differentiation [27,29]. However, much remains to be reconciled between the two studies with respect to the diversity of the genomic EKLF-binding locations and the deduced EKLF regulatory networks. 

Besides EKLF, there are several other factors that have been shown to regulate erythropoiesis [2,5]. In particular, the T-cell Acute Lymphocytic Leukemia 1 (TAL1), also known as the Stem Cell Leukemia (SCL) protein, plays a central role in erythroid differentiation as well. The role of *Tal1*/Scl in primitive erythropoiesis has been demonstrated by the lethality of *Tal1*^−/−^ mice at E9.5 [30,31]. Studies using erythroid cell lines [32] or adult-stage conditional *Tal1* gene knockout mice [33,34] have shown the requirement of *Tal1* in definitive erythropoiesis. Another transcription factor known to play important roles in erythropoiesis is the zinc-finger DNA-binding protein GATA1, which recognizes the consensus binding box ((T/A)GATA(A/G)) [35,36,37]. The cooperative functioning of TAL1 and GATA-1 in the regulation of erythroiepoiesis is closely associated with their physical associations at thousands of genomic loci [38]. Interestingly, *Eklf* appears to be a downstream target gene of the TAL1 factor. Whole-genome ChIP-seq analysis has identified the binding of the TAL1 protein on the *Eklf* promoter in primary fetal liver erythroid cells [39]. Furthermore, in the *Eklf* gene promoter, the composite sequence of GATA-E box-GATA exists, which is a potential binding site of the GATA1-TAL1 protein complex required for the expression of the Eklf gene in a transgenic mouse system [40]. 

In the study reported below, we combined a promoter-based ChIP-chip technique using a high-specificity anti-EKLF antibody and microarray-based gene expression profiling to provide a genome-wide overview of the genes targeted by EKLF in the E14.5 mouse fetal liver cells. Remarkably, *Tal1* has turned out to be a direct target gene of EKLF, indicating the existence of a positive feedback loop between *Eklf* and *Tal1* for the regulation of erythropoiesis in mammals.

## 2. Result

### 2.1. Genome-Wide Identification of EKLF Target Genes by Microarray Hybridization and Promoter-Based ChIP-Chip Analyses

We first carried out a gene profiling analysis to identify the genes regulated by EKLF. The microarrays were hybridized with cDNAs derived from the E14.5 fetal liver RNAs of four wild-type (WT) and four *Eklf* knockout (KO or *Eklf*^−/−^) embryos (Figure 1A). Overall, there were 6975 genes with differential expressions levels between the WT and KO mice (Figure 1B–D). Notably, the confidence index of the microarray hybridization analysis was approximately 70%, as the upregulation/downregulation of 8 out of 12 genes could be validated by semi-quantitation RT-PCR analysis (data not shown).

We then performed a ChIP-chip analysis (Figure 1C) using a promoter-based microarray and the high-specificity polyclonal anti-mouse EKLF antibody (anti-AEK) [19,26]; (Figure 1). The probes on the ChIP-chip array were grouped into 21,536 sequence IDs (SEQ_ID). The promoter of each annotated gene was defined as the region from −3.75 kb upstream to 0.75 kb downstream of the transcription start site (TSS). The SEQ_IDs with at least one significant peak were defined as the potential target binding sites of EKLF. Overall, enriched EKLF-binding was present in 5323 SEQ_IDs, corresponding to 4578 promoters (Figure 1C,D). We also validated the ChIP-chip data by ChIP-qPCR; 9 of 13 promoters on the ChIP-chip list were bound with EKLF, as shown by this assay (data not shown).

The data from the ChIP-chip and the microarray gene profiling experiments were then combined to identify the putative EKLF target genes. After matching between the 11,549 differentially expressed probe sets from the microarray data and the 5323 significant SEQ_IDs from the ChIP-chip array data, 2391 SEQ_IDs (11.1%) from the ChIP-chip data and 3467 probe sets (7.7%) from the microarray hybridization data remained. In the end, the combination of the two data sets resulted in 2644 distinct genes (Figure 1C and Appendix A). This gene list included only genes with an altered expression level in the *Eklf*^−/−^ fetal liver at the *p* < 0.05 level, and with at least one statistically significant EKLF-binding site (*p* < 0.0017) in the promoter region, without considering the fold change of expression and enrichment of binding.

Upon filtering with the effect sizes of the ChIP-chip data (>0.25) and microarray profiling data (>0.5), the number of EKLF-bound and regulated targets was reduced from 2644 to 1866, among which 1156 were down-regulated and 710 were up-regulated in the E14.5 WT fetal liver (Figure 1C,D, Appendix A). The above data support the scenario that the promoter-bound EKLFs could function as either a repressors or activators in vivo. Notably, the promoters bound with and regulated by EKLF were distributed throughout the mouse genome, with no obvious preference for any chromosome (Figure 2A,B).

### 2.2. Functional and Pathway Analysis of EKLF Target Genes

Previous reports by others using the Ingenuity Pathway analysis (IPA) and GeneGo MetaCore analysis platform showed that EKLF target genes were associated with a variety of cellular activities or pathways, including general cellular metabolism, cell maintenance, cell cycle control, DNA replication, general cell development, and the development of a hematologic system [28,41]. To gain further insight into the potential biological roles and functions of EKLF, we applied the IPA software for the analysis of the putative 1866 direct target genes of EKLF. The analysis identified the top five over-represented networks of the down-regulated EKLF targets and the up-regulated EKLF targets, respectively (Appendix A). The significance of the relevant networks was strengthened with use of the higher cut-off score of 25 to ensure that reliable functional networks built by IPA were eligible (Appendix A and Appendix A). Our network analysis confirmed the previously established association of the hematological system development/function with the up-regulated EKLF targets [28]. Notably, the top network associated with either the down-regulated EKLF targets or up-regulated EKLF targets was related to the metabolism and small molecule biochemistry (Appendix A). Additionally, the significant networks/functions associated with the down-regulated EKLF targets were more broad than those with the up-regulated EKLF targets (Appendix A). Overall, our analysis was consistent with previous studies [27,28], in that the promoter occupancy by EKLF also played important roles in the developmental processes, other than erythropeisis and hematological development.

We also used IPA software to group the number of EKLF targets according to their respective biological functions, regulatory pathways, and physiological functions (Appendix A and Appendix A). Of the top five molecular and cellular functions, cell death/survival, cellular assembly/organization, and cellular function/maintenance were overrepresented in both the up-regulated and down-regulated EKLF targets. Further enrichment analysis of the canonical pathways using the IPA software revealed significant overrepresented pathways across the same two genes lists (Appendix A and Appendix A). The prominent enrichment of the up-regulated EKLF targets was related to cell cycle control of the chromosomal replication, EIF2 signaling, mitochondrial dysfunction, hypusine biosynthesis, and tryptophan degradation III (Eukaryotic; Appendix A). The enrichment of the down-regulated EKLF targets was related to insulin receptor signaling, chondroitin sulfate degradation (Metazoa), gap junction signaling, nitric oxide signaling in the cardiovascular system, and PDGF signaling (Appendix A). Together, the above further established the specific functions and associated biological pathways associated with the EKLF target genes.

### 2.3. Identification of Potential Transcription Factors Co-Regulating the EKLF Targets

As co-occurrences of specific transcription factor-binding motifs in the promoters would suggest the cooperation of these factors in transcriptional regulation [42,43], we searched factor-binding motifs across the EKLF-bound promoters, as described in the Material and Methods. Specifically, the consensus transcription factor-binding motifs were ranked based on how often a particular motif occurred within the sequence ID. This observed frequency was applied to all the consensus transcription factor-binding motifs identified within the EKLF-bound regions on each sequence ID (Appendix A and Appendix A). As expected, the most abundant transcription factor-binding motif in the EKLF-bound and regulated promoters was the consensus EKLF-binding sequence CACCC, which was present a total of 2390 times in 2391 of the EKLF target sequence IDs, corresponding to 2143 times in 2644 distinct gene promoters. Consistent with Tallack et al. [27], the binding motifs of known transcription factors functionally interacting with EKLF, such as TAL1 and GATA1 [27], were also identified, which were present at least once in 2390 (2143 distinct gene promoters) and 2384 (2139 distinct gene promoters) of the EKLF target sequence IDs, respectively. In addition, the binding motifs of a number of other transcription factors possibly interacting with EKLF functionally, such as PEA3, LVa, H4TF-1, and XREbf were also identified in this way (Appendix A and Appendix A). 

To investigate the functional cooperation between TAL1 and EKLF or between GATA1 and EKLF, we further analyzed the distance between the binding motifs of GATA1 or TAL1 and that of EKLF. Indeed, the distribution of the TAL1 binding motifs had the highest frequencies between +100 bp and −100 bp from the EKLF binding motifs, indicating a functional cooperation between TAL1 (or possibly Ldb1 complex) and EKLF. Moreover, this cooperation likely acts through the binding of TAL1 upstream of the EKLF protein (Figure 2C). The distribution pattern of the GATA1 binding motifs also supported the cooperation between this factor and EKLF (Figure 2D), although there was no obvious upstream/downstream preference between these two factors.

### 2.4. Likelihood of Tal1 Gene as a Regulatory Target of EKLF in E14.5 Fetal Liver Cells

Our motif analysis across the EKLF-bound promoters revealed that the binding motifs of the transcriptional factor TAL1 had the highest frequency of co-occupancy with the binding motifs of EKLF (Appendix A). Interestingly, in the hematopoietic system, the transcription factor duet EKLF-GATA1 or TAL1-GATA1 served as part of the specific activation complex(es) in the erythroid cells [43,44,45], and both the *Eklf* gene [46,47] and the *Tal1* gene [48,49] were activated by the GATA1 factor [28,29]. We thus further investigated whether there was also an epistatic relationship between *Eklf* and *Tal1*. 

Gene expression profiling by microarray hybridization revealed a 2.5-fold (effect size = 1.3139) down-regulation of the *Tal1* transcript in the E14.5 *Eklf*^−/−^ fetal liver in comparison with the wild-type E14.5 fetal liver (Appendix A). This microarray data were validated by RT-qPCR. As shown, the level of *Tal1* mRNA in the *Eklf*^−/−^ fetal livers was decreased significantly, down to 47% of the level detected in wild-type fetal liver (left histograph, Figure 3A). In parallel, the TAL1 protein level in the *Eklf*^−/−^ fetal livers cells was also down-regulated by 70% when compared with the wild type (right panels and histograph, Figure 3A). Thus, not only the TAL1 factor could activate the *Eklf* gene transcription [39,40], but the *Tal1* gene might also be a regulatory target of EKLF. As *Eklf*^−/−^ mice and *Tal1*^−/−^ mice both exhibited a deficit of erythroid-lineage cells after the stage of basophilic erythroblasts [33,46], we suspected that the promotion of erythroid terminal differentiation from pro-erythroblasts to basophilic erythroblasts very likely required EKLF-dependent activation of the *Tal1* gene transcription. 

### 2.5. EKLF As an Activator of Tal1 Gene Expression during Erythroid Differentiation

To further examine whether EKLF was an activator of *Tal1* gene transcription in erythroid cells, we first analyzed the expression level of *Tal1* mRNA in cultured mouse erythroid leukemic (MEL) cells during DMSO induced erythroid differentiation. Similar to βmaj mRNA, the *Tal1* mRNA was expressed in un-induced MEL at a basal level, which was increased by 2–3 fold upon DMSO differentiation (top, Figure 3B). Consistent with this, the protein level of TAL1 was also up-regulated during a 48 h period of DMSO-induced differentiation, but was down-regulated subsequently (bottom, Figure 3B). The up-regulation of the *Tal1* gene supported the scenario that a sustained higher expression of the *Tal1* gene was required for erythroid differentiation. The biphasic expression profile of the TAL1 protein further suggested that the requirement of TAL1 for MEL cell differentiation was up to 48 h after DMSO-induction, which corresponded to the basophilic/polychromatic stages of erythroid differentiation. 

We then analyzed the *Tal1* mRNA levels in two independent MEL cell-derived stable clones, 4D7 and 2M12. As shown in Figure 3C, the knock-down of *Eklf* mRNA by the doxycycline-induced shRNAs led to a significant reduction in the *Tal1* mRNA under the condition of DMSO-induced erythroid differentiation, but not in the un-induced MEL cells. The latter result further supported that EKLF was not part of the regulatory program of *Tal1* gene transcription in MEL cells prior to their differentiation. The data of Figure 3C indicate that EKLF was required for the activation of *Tal1* gene transcription during the DMSO-induced erythroid differentiation of MEL cells. Together with the loss-of-function of *Eklf* studied in the mouse fetal liver (Figure 3A), we conclude that while TAL1 is a known activator of *Eklf* gene transcription, EKLF also positively regulates *Tal1* gene transcription during erythroid differentiation from CFU-E/ pro-erythroblasts to the basophilic/polychromatic erythroid cells.

### 2.6. Binding In Vivo of EKLF to the Upstream Promoter of Tal1 Gene

How would EKLF activate the *Tal1* gene transcription during erythroid differentiation? It could either directly activate the *Tal1* gene through DNA-binding in the regulatory regions of the gene, e.g., its promoter or enhancer, or indirectly through other transcriptional cascades. In an interesting association with the above data of *Tal1* expression in the presence and absence of EKLF, the ChIP-chip analysis identified two regions with significant reads of EKLF-binding, one of which (region I, 114,551,700–114,552,900 on chromosome 4, NCBI 36/mm8) was located around the *Tal1* gene in the E14.5 fetal liver cells (Figure 4A, Appendix A, Appendix A). In the mouse erythroid cells, the *Tal1* gene encodes a *Tal1* mRNA isoform A (Figure 4B) consisting of five exons, with the most upstream exon1 located at 115,056,426–115,056,469 (NCBI 36/mm10). However, no CCAAT box or TATA box or CACCC box could be found 300 bp upstream of this exon 1. Instead, we found these motifs in a region ~860 bp upstream of isoform A exon 1 (Figure 4B,C; see also sequence in Figure 5A). We thus suspected that exon 1 of the *Tal1* gene might be longer than currently documented in the database. Alternatively, there might be another exon upstream of the exon 1 of isoform A. 

To solve the issue, we carried out a semi-quantitative RT-PCR analysis of the MEL cell RNAs using different sets of primers. As shown in the bottom panels of Figure 4B, the use of the forward primer PF-3 with any one of four different reverse primes (AR-1, AR-2, AR-3, and AR-4) would not generate a RT-PCR band on the gel. On the other hand, the use of the forward primer PF-1 or PF-2 together with the four reverse primers generated RT-PCR bands, the lengths of which were consistent with the existence of an exon (115,055,766–115,056,469, NCBI 36/mm10) consisting of the previously known isoform A exon 1 at its 3′ region (diagram, Figure 4B). Based on these RT-PCR data and the common distance (25–27 bp) between the promoter TATA box and transcription start site(s) of the polymerase II-dependent genes, we suggest a map of the promoter region of *Tal1* gene upstream of the newly identified exon1, which contains the TATA box at −28, two CCAAT boxes at −133 and −57, and three CACCC boxes (−788, −710, and −185) upstream of the transcription start site or TSS (Figure 4C and Figure 5A). 

To validate the in vivo binding of EKLF in the newly identified *Tal1* promoter, we carried out a ChIPq-PCR analysis. As shown in Figure 4C, use of four different sets of primers spanning different regions upstream and downstream of the *Tal1* transcription start site (TSS) indicated EKLF-binding to region b containing the distal CACCC boxes E1 at −788/E2 at −710, and to region c containing the proximal CACCC box E3 at −185. 

### 2.7. Binding In Vivo of EKLF to the Proximal CACCC Box of Tal1 Promoter-Genomic Footprinting Analysis 

In order to examine whether EKLF indeed bound to the proximal CACCC box of the *Tal1* promoter in differentiated erythroid cells, we next carried out a genomic footprinting assay of the *Tal1* promoter in MEL cells before and after DMSO induction (Figure 5). As shown, upon DMSO induction of the MEL cells, genomic footprints appeared at the distal CACCC box E1 and more prominently the proximal CACCC box E3 at −185 (Figure 5). On the other hand, the distal CACCC box E2 at −710 was not protected in MEL cells with or without DMSO induction. Notably, the intensities of gel bands at −133, −132, −57, and −56 appeared to be enhanced upon DMSO induction, suggesting a binding of factor(s) at the two CCAAT boxes as well (Figure 5). These genomic footprinting data support the scenario that EKLF positively regulates the *Tal1* promoter activity through binding mainly to the proximal promoter CACCC box E3. This would facilitate the recruitment of other factors, including the CCAAT box-binding protein(s) to the *Tal1* promoter. 

### 2.8. Requirement of the Proximal CACCC Motif for Transcriptional Activation of the Tal1 Promoter by EKLF

To investigate whether EKLF was indeed an activator of *Tal1* gene transcription through binding to the proximal CACCC promoter box, we constructed a reporter plasmid p*Tal1*-luc, in which the *Tal1* promoter region from −900 to −1 was cloned upstream of the luciferase reporter. Three mutant reporter plasmids, p*Tal1*(Mut E1)-Luc, p*Tal1*(Mut E2)-Luc, and p*Tal1*(Mut E3)-Luc, were also constructed, in which the CACCC box E1, E2, or E3 was mutated (Figure 6A). Human 293T cells were then co-transfected with one of these four reporter plasmids, plus an expression plasmid pFlag-EKLF. As shown in Figure 6B, the luciferase reporter activity in the cells co-transfected with p*Tal1*-Luc, p*Tal1*(Mut E1)-Luc, or p*Tal1*(Mut E2)-Luc increased in a Flag-EKLF dose-dependent manner. However, mutation at the E3 box of the reporter plasmid p*Tal1*(Mut E3)-Luc prohibited this increase. This result, in combination with the genomic footprinting data of Figure 5, demonstrate explicitly that the binding of EKLF to the proximal CACCC box E3, but not the distal E1 or E2 box, in differentiated erythroid cells is required for the transcriptional activation of the *Tal1* promoter.

## 3. Discussion

A well-coordinated group of transcription factors regulate similar or distinct sets of target genes, which build up the diverse functional networks and biological pathways governing the process of erythropoiesis. Among these factors are GATA1, FOG1, FLI1, PU.1, TAL1/SCL, and EKLF [2,3,4,10]. Previously, global analyses by gene expression profiling with the use of the microarrays have suggested the potential target genes and genetic pathways that function in erythropoiesis, as regulated by GATA1, TAL1/SCL, and EKLF [17,18,29,36,39,49]. Later, ChIP analysis in combination with next-generation sequencing and microarray hybridization further provided lists of genes that could be regulated directly, through DNA-binding, by these factors [27,28,38,39,50]. Among the factors the potential regulatory targets of which have been studied globally is EKLF. In particular, the two sets of ChIP-Seq analyses have each provided a set of direct target genes of EKLF in the mouse fetal liver cells [27,28]. The change of binding of EKLF to its potential gene targets during differentiation from erythroid progenitors to erythroblasts in the E13.5 fetal liver has also been analyzed [28]. However, these two studies have displayed divergent data with respect to the identities of genes directly regulated by EKLF. 

In this study, we analyzed the regulatory functions of EKLF in E14.5 mouse fetal liver cells through the combined use of genome-wide expression profiling and a promoter ChIP-chip assay. Unexpectedly, the number of direct gene targets (1866), as defined by the occupancy of EKLF within −3.75 kb to +0.75 kb relative to TSS (1.2-fold enrichment) and a >1.4 fold change in the expression levels upon depletion of *Eklf* in the gene knockout mice, are significantly higher than those derived from Tallack et al. [27] and Pilon et al. [28]. As shown in Appendix A, of the 1866 EKLF target genes that we identified, 257 genes (13.7%) overlapped with the data set from Tallack et al. [27], and 231 genes (12.3%) overlapped with the data set from Pilon et al. [28]. Furthermore, the number of overlapping genes between those two data sets was only 199. Moreover, among the direct targets identified in the three studies, only 55 (2.9% of 1866) were in common (Appendix A). The inconsistencies of the conclusions among the three groups with respect to the direct target genes of EKLF likely resulted from the use of different antibodies; different approaches; different developmental stages of the embryos analyzed; different mouse strains; different cell types; and, finally, analyses using different peak calling methods. Moreover, we used 1.4-fold as the cutoff line, rather than 2-fold chosen by the other two groups [17,28,29], when comparing the WT and *Eklf*^−/−^ expression profiles. This lower cutoff line may have allowed us to find more candidate targets that display subtle expression differences, but have a prominent functional significance. Notably, the use of a higher cut-off line, i.e., 2-fold instead of 1.4-fold, reached a similar conclusion (Appendix A). 

One surprising outcome of our genome-wide study is the existence of a positive feedback loop between the two well-known erythroid-enriched transcription factors, EKLF and TAL1, in early erythroid differentiation. By loss-of-function analysis, we show that EKLF also positively regulates the expression of *Tal1* during erythroid differentiation (Figure 3). In particular, the induced depletion of *Eklf* drastically lowers the expression level of *Tal1* in DMSO-induced MEL cells (Figure 3C). The combined data from the ChIP-chip, genomic footprinting, and transient reporter assays further indicate that EKLF activates the *Tal1* gene transcription through binding to the proximal CACCC box in the newly identified *Tal1* promoter (Figure 4 and Figure 5). Consistent with this scenario of mutual activations of *Tal1* and *Eklf*, the mRNAs of *Tal1* and *Eklf* were both progressively up-regulated during erythroid differentiation of the primary mouse fetal liver cells [51]. Thus, our finding of the positive regulation of the *Tal1* gene by EKLF demonstrates the existence of a *Tal1*-*Eklf* positive feedback loop that promotes the mammalian erythroid differentiation in a tightly regulated time window, from the transition of Pro-E to Baso-E of the erythroid lineage.

We propose the following scenario for the mutual activation of *Tal1* and *Eklf*, as well as the functional consequences of this positive feedback loop during erythroid differentiation. In the erythroid lineage at the BFU-E/CFU-E/Pro-E stages, the two factors are already expressed at basal levels. EKLF is retained by FOE in the cytosol [26], while the TAL1 protein positively regulates the expression of *Eklf* [38,39]. When the cells enter the Baso-E stage, the EKLF protein is released from its physical interaction with FOE in the cytoplasm and is imported into the nucleus [26]. The imported EKLF binds to the E3 box of the *Tal1* promoter to enhance the promoter activity of *Tal1* (Figure 5 and Figure 6). This positive feedback loop rapidly amplifies both factors during erythroid terminal differentiation. As a result, the EKLF-mediated activation of *Tal1* may act as a valve that facilitates the commitment of the erythroid lineage from MEP through promoting the differentiation transition from Pro-E to Baso-E, thus sustaining the process after Baso-E. Furthermore, there is a high frequency of co-occupancy of EKLF and TAL1 in a number of promoters that are active in erythroid cell lines or erythroid tissues (Appendix A) [27,28,29]. Thus, the *Eklf*/*Tal1* loop would irreversibly promote erythroid terminal differentiation through the up-regulation of not only *Eklf* and *Tal1*, but also their mutual downstream targets that are crucial for erythroid differentiation. For the latter process, the EKLF and TAL1 proteins may work within the same transcriptional complex(es) that bind to the composite CACCC box-E box in the promoters of these downstream targets [45]. The positive feedback regulatory loop between EKLF and TAL1, as identified in this study, provides a mechanism ensuring the commitment to erythroid differentiation among the multiple lineages of the hematopoietic system.

## 4. Material and Methods

### 4.1. Generation of Eklf^−/−^ Mice

As described elsewhere [11], the generation of B6 mouse lines with homozygous knockout of the *Eklf* gene, *Eklf*^−/−^, was carried out in the Transgenic Core Facility (TCF) of IMB, Academia Sinica, following the standard protocols with the use of the BAC construct containing genetically engineered *Eklf* locus and E2A-Cre mice.

### 4.2. Gene expression Profiling by Affymetrix Array Hybridization

The E14.5 mouse fetal livers from WT and *Eklf*^−/−^ mouse fetuses were homogenized by repeated pipetting in phosphate-buffered saline (PBS) (10 mM phosphate, 0.15 M NaCl [pH 7.4]). The total RNAs were then isolated with Trizol reagent (Invitrogen, Carlsbad, CA, USA) and were subjected to genome-scale gene expression profiling using the Mouse Genome Array 430A 2.0 (Affymetrix, Inc., Santa Clara, CA, USA). The standard MAS5.0 method was applied to normalize the gene expression data. The gene expression values were log-transformed for later comparative analysis. Statistical analysis was carried out using R 3.0.2 language (R Development Core Team, 2013, http://www.R-project.org, accessed on 21 July 2021)

### 4.3. Identification of Differentially Expressed Genes

The genes with differential expression patterns between the WT and *Eklf* ^−/−^ mice E14.5 fetal liver were first identified using a two-sided two sample *t*-test with the significance level at 0.05. As the set of probes for each annotated gene should all exhibit the same direction or sign when comparing the WT and *Eklf*^−/−^ samples, this consistency check was used to remove 257 ambiguous genes from the gene list. After filtering by the *p*-value threshold, a subset containing 12,277 statistically significant probe sets was obtained. 

### 4.4. Identification of EKLF-Bound Targets by Using NimbleGen ChIP-Chip Array Hybridization

The E14.5 mouse fetal liver cells were cross-linked, sheared, and the EKLF bound-chromatin complexes were immuno-precipitated (ChIP) with the AEK antibody [20] and rabbit IgG, respectively. The DNA was then purified from the immunoprecipitated chromatin samples using a QIAquick PCR purification kit (Qiagen, Hilden, Germany) and amplified by the Sigma GenomePlex WGA kit for hybridization with the Roche NimbleGen Mouse ChIP-chip 385 K RefSeq promoter arrays. 

There were 768,217 probes on the NimbleGen 385 K ChIP-chip array. These probes were grouped into 21,536 sequence IDs, each of which contained 5 to 320 probes that ranged from 49 bp to 74 bp in length. The distances between the probes in the same sequence ID ranged from 100 bp to 3700 bp. In general, these sequence IDs are located in the promoter regions of the genes, roughly from −3.75 kb to +0.75 kb, relative to the transcription start site (TSS). The sequence ID was assigned a gene name when the gene’s coding sequence overlapped with the region from 10 kb upstream to 10 kb downstream of the sequence ID. In this way, 652 sequence IDs were found to be located in the intergenic regions and 20,884 sequence IDs were near the coding regions of the genes.

To identify the binding targets of EKLF, the moving window with a size equal to 5 was adopted to test the hypothesis on a positive mean value using a one-sided *t*-test with the significant level set at 0.0017. This smaller cut-off value was chosen to account for the multiple comparisons. Specifically, there were 35 probes in each sequence ID, and the adjusted *p*-value was derived by (1 − (1 − 0.0017)^30^) ~0.05. The moving window was applied to each sequence ID separately. Finally, the results were summarized at the sequence ID level, and a sequence ID would be defined as a target site of EKLF if there was a significant peak in the sequence ID.

### 4.5. Matching between Affymetrix Probes and NimbleGen ChIP-Chip Probes

The E14.5 fetal liver gene expression data obtained from the Affymetrix array hybridization analysis allowed us to further reduce the false positives from the ChIP-chip dataset. To do this, the annotation strategy used in annotating the sequence IDs in the ChIP-chip array was adopted to match the probes from these two platforms by gene symbols. After this procedure, there were a total of 78,634 matched pairs between the ChIP-chip sequence IDs and Affymetrix probes.

### 4.6. Co-Occurrence of Binding Motifs and Relative Distance Distribution

First, 226 known transcription factor-binding motifs were extracted from the previous report [28]. For each of these binding motifs, the number of sequences in the mouse genome bearing the motif was sorted. The top co-existing binding motifs with EKLF were then further investigated. The relative distance between a co-existing motif and the EKLF-binding motif was calculated for each sequence ID. However, when multiple binding motifs existed in the same sequence ID, multiple distances would be generated. In that case, the shortest distance was selected as the representative distance in that sequence ID. The relative distance distribution was then plotted to inspect the potential localization biases. The existence of a localization bias provided further indication that two transcription factors might interact in certain way to regulate the particular target gene(s).

### 4.7. Functional Enrichment Analysis

An analysis was carried out with the use of IPA (Ingenuity^®^ Systems, www.ingenuity.com, accessed on 21 July 2021) to identify the genes significantly associated with specific biological functions and/or diseases in the Ingenuity Knowledge Base. Right-tailed Fisher’s exact test was used to calculate the *p*-value, determining the probability that each biological function and/or disease assigned to that data set was due to chance alone. The list of genes with significant EKLF-binding enrichment and deemed to be expressed differentially in WT and KO mice fetal livers was imported into IPA. The up-regulated and down-regulated EKLF targets were first mapped to the functional networks available in the IPA database, and then ranked by scores computed with the right-tailed Fisher’s exact test mentioned above. As listed in Appendix A, this analysis identified significant over-represented molecular and cellular functions (*p* value < 0.05) associated with the imported up-regulated and down-regulated EKLF targets that were eligible (score > 25), with significance scores of 26 and 34, respectively (Appendix A). 

### 4.8. ChIP-qPCR 

The ChIP-PCR analysis followed the procedures of Daftari et al. [52]. The sonicated cell extracts from formaldehyde cross-linked E14.5-day mouse fetal liver cells were immuno-precipitated with anti-EKLF and purified rabbit IgG, respectively. The precipitated chromatin DNA was purified and analyzed by quantitative PCR (qPCR) in the Roche LightCycle Nano real-time system. The sequences of the primers used for q-PCR designed by our lab are listed in Appendix A. Each target gene was amplified with one set of primers flanking the putative EKLF-binding CACCC motif(s) and two sets of non-specific primers bracketing the regions located upstream and downstream of the CACCC motif(s), respectively. 

### 4.9. Plasmid Construction

Mouse *Eklf* cDNA was derived by the RT-PCR of RNA from DMSO-induced MEL cells and was cloned into the vector pCMV-Flag (Invitrogen), resulting in pFlag-EKLF. Plasmids for the luciferase reporter assay were constructed in the following way: *Tal1* promoter region from −1 to −900 relative to the transcription start site of the newly identified *Tal1* exon 1 was amplified by PCR of mouse genomic DNA, with the addition of a XhoI cutting site at the 5′ end and a HindIII site at 3′ end, and cloned into the XhoI and HindIII sites in the psiCHECK™-2 Vector (Promega, Madison, WI, USA), resulting in the plasmid p*Tal1*-Luc. *Tal1* promoter DNA fragments with the putative EKLF-binding CACCC box(es) mutated were generated by fusion PCR, using the endogenous *Tal1* promoter as the template. The sequences of the three mutated CACCC boxes and their flanking regions in these fragments were E1 box, 5′-CAGGCAAAACCAGGGACCAcatatTTAAAAATGATTCCCCTTCTCAAG-3′; E2 box, 5′-CAATAGCTCTTCAGTTAGCGGTGAAGGCTCATGAAcatatCCAC-3′; and E3 boxes, 5′-GAGTTATTGACACAGCCCTGTcatatCCTCCCCCCACTG-3′. The inserts of all of the plasmids were verified by DNA sequencing before use.

### 4.10. Cell Culture, Differentiation, DNA Transfection, and Knockdown of Gene Expression

The murine erythroleukemia cell line (MEL) was cultured in Dulbecco’s modified Eagle medium containing 20% fetal bovine serum (Gibco, Carlsbad, CA, USA), 50 units/mL of penicillin, and 50 μg/mL of streptomycin (Invitrogen). For the induction of differentiation, the cells at a density of 5 × 10^5^/mL were supplemented with 2% dimethyl sulfoxide (DMSO; Merck) and the culturing was continued for another 24 to 72 h. DNA transfection of the MEL cells and K562 cells was carried out using the TurboFect transfection reagent (Thermo Scientific, Waltham, MA, USA) and Lipofectamine^®^ 2000 transfection reagent (Life Technologies, Carlsbad, CA, USA), respectively. 

For knockdown of the *Eklf* gene expression, MEL cell line-derived clones 4D7 and 2M12 [7] were maintained in 20 μg/mL of blasticidin (Invitrogen) and 1 mg/mL of G418 (Gibco). Differentiation of the 4D7 and 2M12 cells was induced by 2% dimethyl sulfoxide (DMSO; Merck) for 48 h. The expression of shRNA targeting and the knocking-down *Eklf* was induced with the addition of 2 μg/mL of doxycycline (Clontech, Kusatsu, Japan) for 96 h, as described in Bouilloux et al. [7]. 

### 4.11. RNA Analysis

The total RNA from the MEL cells and fetal liver suspension cells were extracted with the TRIzol reagent (Invitrogen). cDNAs were synthesized using SuperScript II Reverse Transcriptase (RT) (Invitrogen) and oligo-dT primer (Invitrogen). Taq DNA polymerase was used for semi-quantitative RT-PCR analysis of the cDNAs. Quantitative real-time PCR (qPCR) analysis of the cDNAs was carried out using the LightCycler^®^ 480 SYBR Green I Master (Roche Life Science, Penzberg, Germany) and the products were detected by a Roche LightCycler LC480 Real-Time PCR instrument. The primers used for the qPCR analysis were designed following previous reports or from the online database PrimerBank: http://pga.mgh.harvard.edu/primerbank, accessed on 21 July 2021. The primers used for validating the microarray data and for *Tal1* exon 1 identification by RT-PCR were designed by our lab. The sequences of the DNA primers used in semi-quantitative RT-PCR and real-time RT-qPCR are available upon request.

### 4.12. Western Blotting Analysis and Antibodies

Whole-cell extract of MEL or mouse fetal liver cells were analyzed by polyacrylamide gel electrophories (PAGE) and Western blotting, following the standard protocols. Enhanced chemiluminescence (ECL) detection system (Omics Biotechnology Co., Taipei, Taiwan) was used to visualize the hybridizing bands on the blots. Goat anti-TAL1 antibodies, sc-12982 and sc-12984, were purchased from Santa Cruz, Inc., Anti-Flag (M2), anti-Tubulin(B-5-1-2), and anti-β-Actin (AC-15) mouse antibodies were purchased from Sigma-Aldrich (St. Louis, MO, USA). The anti-EKLF antibody (anti-AEK) was homemade [19].

### 4.13. Reporter Assay

For the luciferase reporter assay in 293T cells, 1 μg of each of the wild-type p*Tal1*-Luc plasmid or its mutant forms were transfected into 4 × 10^5^ /mL of cells. The total amount of transfected DNA was kept at 0~3 μg, with the addition of 0~3 μg of empty vector pCMV-Flag. After 24 h, the luciferase activities were measured using the Dual-Luciferase^®^ Reporter Assay System (Promega). Firefly luciferase activity was used as an internal control and Renilla activity was used to monitor the transactivity of the *Tal1* promoter and its mutant forms.

### 4.14. In Vivo Genomic Footprinting

The status of nuclear factor-binding in the living MEL cells was investigated by dimethyl sulfate (DMS) cleavage in vivo and ligation-mediated PCR (LMPCR), as described previously [53,54,55,56], with some modification. The distal promoter region was analyzed with primer set D (P1[5′-885 GCTCACAAA CT CCT GTTTCAGAGGAG-860 3′], P2 [5′-867CAGAGGAGCTAATGTT CTGCCTTCTTC-840 3′], and P3 [5′-867 CAGAGGAGCTAATGTGTCTGCCTTCTTCTAG-837 3′]), the central promoter region was analyzed with primer set C (P1[5′-817GACATTAATACAGGCAAAACCAGG GACC-790 3′], P2[5′-798CCAGGGACCAC ACCCTTAAAAATGATTCC-770 3′], and P3[5′-798CCAGGGACCAC ACC CTTAAAAATGATTCCCC-768 3′]); and the proximal promoter region was analyzed with primer set P(P1[5′ 320GAAAGAAAAACCCAG A TACTCCTCAGC-294 3′], P2 [5′-287GGTTCCTACAATGTACCTATGG GCTTC-261 3′], and P3[5′-287GG TTCCTACAATGTACCTATGGGCTTCAATG-257 3′]). Different batches of DMS-treated cells were analyzed several times to check for consistency of the protection patterns. The relative intensities of the bands on the autoradiographs were estimated in an AlphaImager 2200 (Clontech).

## Figures and Tables

**Figure 1 ijms-22-08024-f001:**
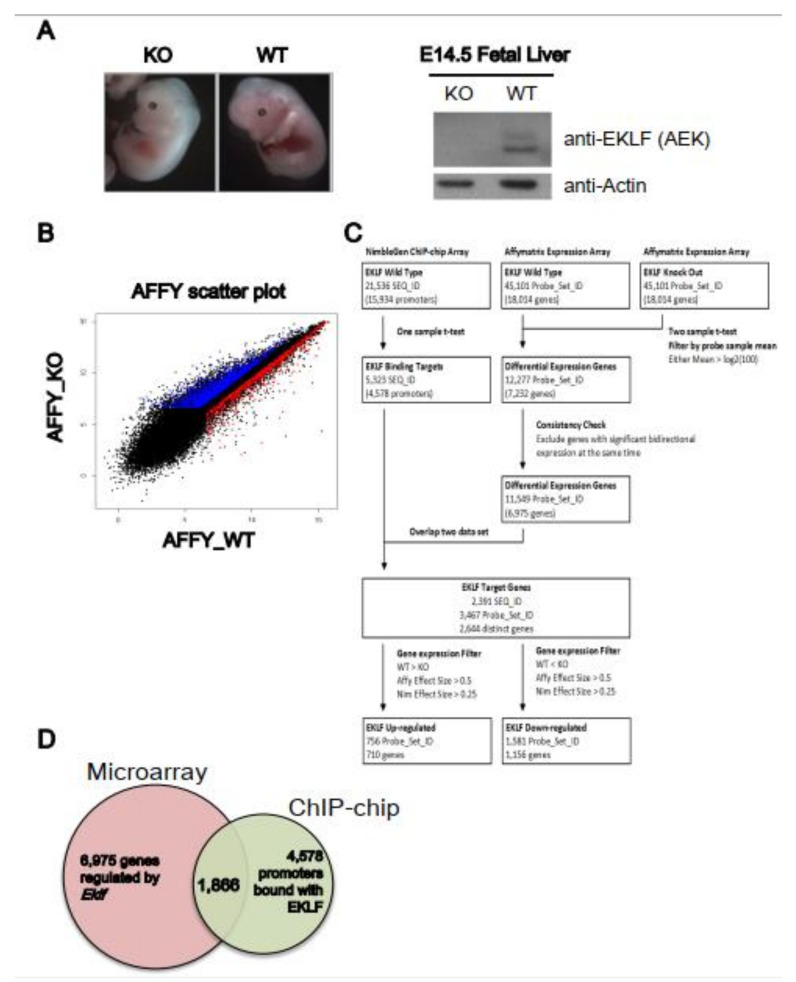
Identification of EKLF target genes by global gene expression profiling. (**A**) Left panels, representative appearance of E14.5 embryos of wild-type and *Eklf*^−/−^ mice. Right panels, Western blotting patterns of EKLF protein in E14.5 fetal lives. Actin was used as the gel loading control. (**B**) Scatter plot comparing the gene expression profiles of the E14.5 fetal liver cells of WT and *Eklf*^−/−^ mice by Affymatrix array hybridization. Each gene on the arrays is displayed as a single dot on a logarithmic (log2) graph. The genes up-regulated and down-regulated by EKLF are indicated by the red and blue dots, respectively. (**C**) Overview of the workflow of ChIP-chip and microarray expression profiling. The flow chart illustrates the procedures used for the analysis of the NimbleGen promoter ChIP-chip data and Affymatrix differential expression profiling data. The number of genes (SEQ_ID or Probe_Set) after data processing at each step is indicated in parentheses. (**D**) The Venn diagram showing the overlapping between gene sets derived from the microarray hybridization analysis (6975 genes) and ChIP-chip analysis (4578 genes) of the E14.5 fetal liver cells of WT and *Eklf*^−/−^ mice, respectively.

**Figure 2 ijms-22-08024-f002:**
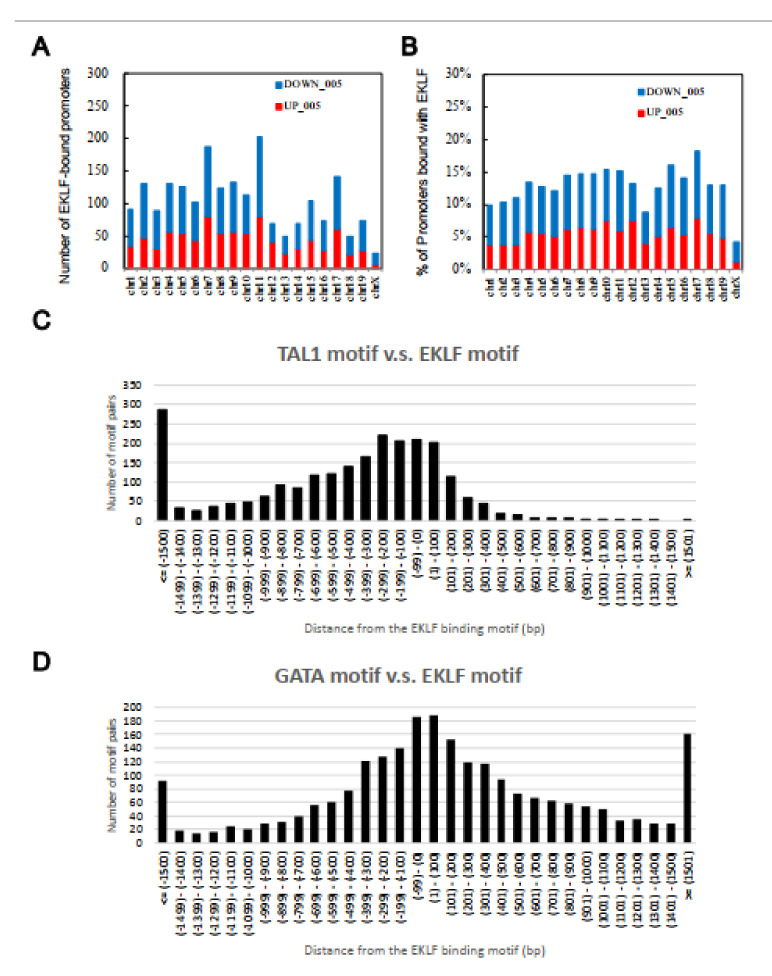
Chromosome distribution patterns from the global analysis of direct target genes of EKLF in the E14.5 fetal liver cells. (**A**) The number of putative EKLF-bound promoters on the different mouse chromosomes. (**B**) The percentage of promoters of the individual mouse chromosomes bound with EKLF. (**C**,**D**) Distributions of the distances between the binding motif of TAL1 (**C**) and GATA1 (**D**), and that of EKLF on the mouse genome in the E14.5 fetal liver cells. Upstream locations are indicated by the “−” sign.

**Figure 3 ijms-22-08024-f003:**
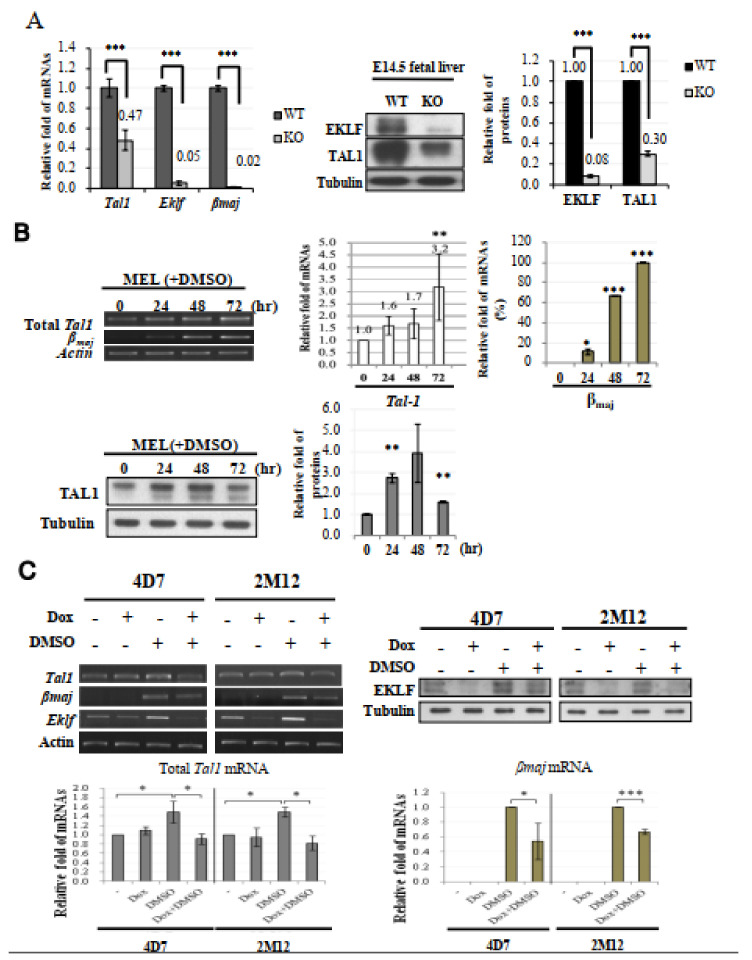
*Tal1* as a direct target gene of EKLF. (**A**) Left, bar diagram of the relative mRNA levels of *Tal1*, *Eklf,* and βmaj in E14.5 fetal liver cells of the WT and *Eklf*^−/−^ (KO) mice, as analyzed by RT-qPCR. *** *p* < 0.001 by *t* test. Error bars, SEM. Middle panels and right histobar diagram, Western blotting analysis of TAL1 and EKLF in the E14.5 fetal livers of WT and *Eklf*^−/−^ (KO) mice. Tubulin was used as the loading control. *** *p* < 0.001 by *t* test. Error bars, STD. (**B**) Top, expression levels of *Tal1* and βmaj in MEL cells without or with DMSO induction for 72 h. The gel patterns of the semi-quantitative RT-PCR bands are shown on the left, and the histographs of the statistical analysis of the data are shown on the right. * *p* < 0.05, ** *p* < 0.01, and *** *p* < 0.001 by *t* test. Error bars, SD. Bottom, Western blotting analysis of the levels of the TAL1 protein in MEL cells during DMSO-induced differentiation. Tubulin was used as the loading control. The statistical analysis of the data is shown in the bar diagrams on the right. ** *p* < 0.01 by *t* test. Error bars, SD. (**C**) Analysis of the gene expression in 4D7 and 2M12 cells without and with doxycycline (Dox)-induced expression of *Eklf* shRNA. The cells without or with induction by DMSO for 48 h were treated with doxycycline. The levels of EKLF protein in the whole cell extracts were then analyzed by Western blotting, as exemplified in the upper right panels. Tubulin was used as the loading control. RT-PCR analysis showed that the knockdown of EKLF reduced the levels of total *Tal1* mRNA and βmaj mRNA in DMSO-induced cells, as exemplified in the upper left panels and statistically analyzed in the two histobar diagrams below. The gel band signals were all normalized to that of actin. * *p* < 0.05, and *** *p* < 0.001 by Student’s *t* test. Error bars, SD.

**Figure 4 ijms-22-08024-f004:**
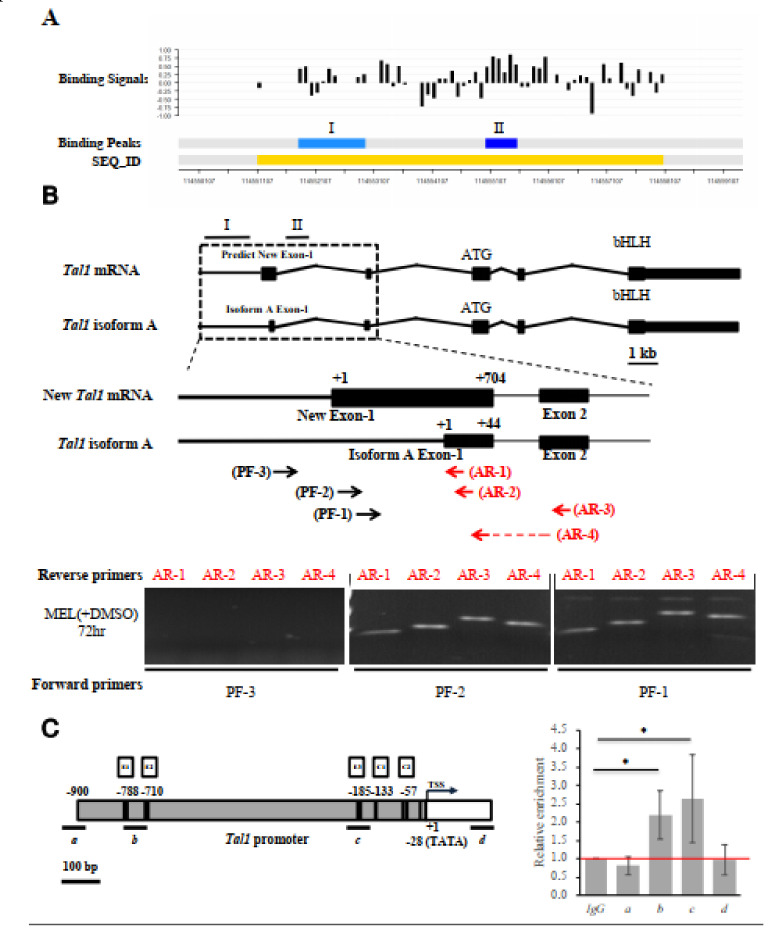
Identification and characterization of the authentic exon-1 and promoter of *Tal1* gene. (**A**) ChIP-chip promoter array data around the *Tal1* gene region. The black bars indicate the signals from the individual probes (“binding signals”). The blue regions I and II indicate the EKLF-binding signals from ChIP-chip promoter array analysis of two different mouse E14.5 fetal liver samples (“binding peaks”). The yellow region indicates the sequence ID (SEQ_ID). (**B**) RT-PCR validation of the newly identified exon 1 of *Tal1* mRNA. Top, maps of the newly identified exon structure of *Tal1* mRNA in comparison with that of *Tal1* isoform A. Exons 1–5 are represented by the black boxes. Middle, maps of the exon-1 of *Tal* mRNA and *Tal1* isoform A, respectively, are shown above the primers used for RT-PCR analysis of the *Tal1* mRNA. The sequences of the reverse primers AR-1 and AR-2 are derived from the exon 1 of *Tal1* isoform A. AR-3 is derived from the exon 2 sequence. The reverse prime AR-4 is across exons 1 and 2. The sequences of the forward primers PF-1 and PF-2 are from the predicted new exon-1, while PF-3 is from the upstream region. Bottom, gel band patterns of the RT-PCR analysis of DMSO and MEL cell RNAs using different sets of PCR primers. (**C**) ChIP-qPCR analysis of EKLF-binding to the *Tal1* promoter. The map of the *Tal1* promoter region is shown on the left, with the CACCC boxes (E1, E2, and E3), CCAAT boxes (C1 and C2), the TATA box, and the transcription start site (TSS, +1) indicated. The four regions (a, b, c, d) are bracketed by the four primer sets used in qPCR analysis of chromatin from DMSO-induced MEL cells immunoprecipitated with anti-EKLF. The relative folds of the enrichment of the chromatin DNA samples pulled down by anti-EKLF are calculated as the Cq values over those derived from use of the IgG and shown in the right histograph. Error bars represent standard deviations from 3~7 biological repeats. The statistical significance of the difference between the experimental and control groups was determined by the two-tailed Student’s *t* test, * *p* < 0.05.

**Figure 5 ijms-22-08024-f005:**
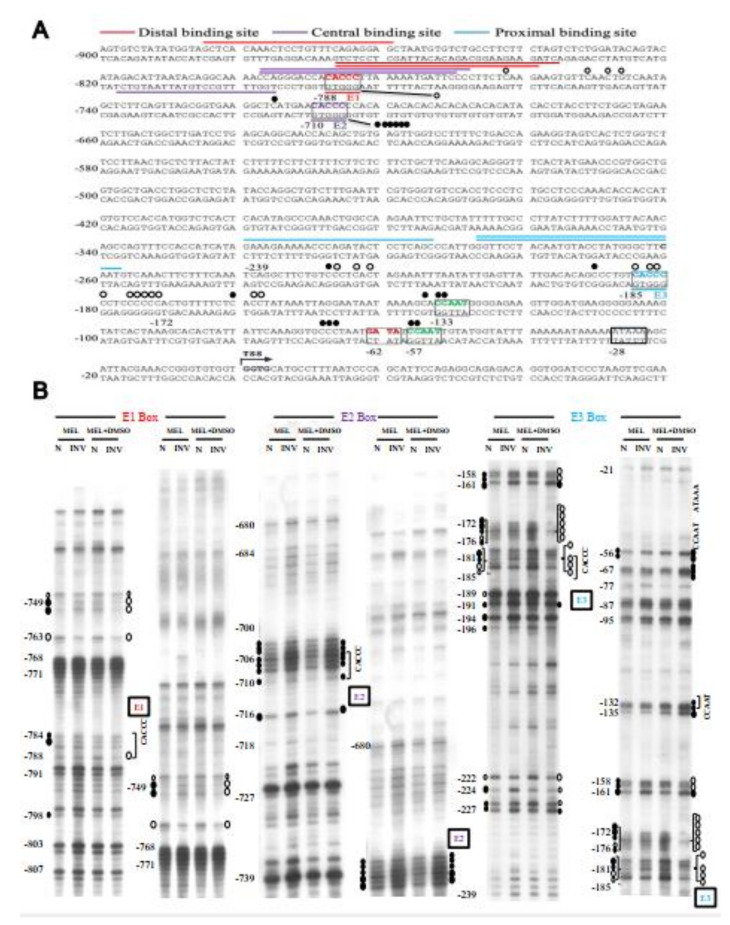
Genomic footprinting analysis of the promoter of *Tal1* gene in MEL cells. (**A**) The protected bases (○) and hyper-reactive bases (●) of the *Tal1* promoter region in MEL cells after DMSO induction, as deduced from the in vivo DMS footprinting analysis, are labeled on the DNA sequence. The footprinting pattern indicates the binding of EKLF on the E3 box. (**B**) The representative autoradiographs of the analysis of the upper strand of the *Tal1* promoter region by in vivo DMS protection and LMPCR assays are shown. Locations of different factor-binding motifs/boxes, i.e., E1, E2, E3, CCAAT, and ATAAA, are indicated on the right of the gel patterns. Numbers on the left correlate with those indicated on the sequence in (**A**). The patterns in the N and INV lanes are the results from the in vitro and in vivo DMS cleavages, respectively. Only those residues consistently showing differences from the controls are indicated. The sizes of the circles reflect the different extents of the protection or enhancement of DMS cleavage in vivo vs. in vitro.

**Figure 6 ijms-22-08024-f006:**
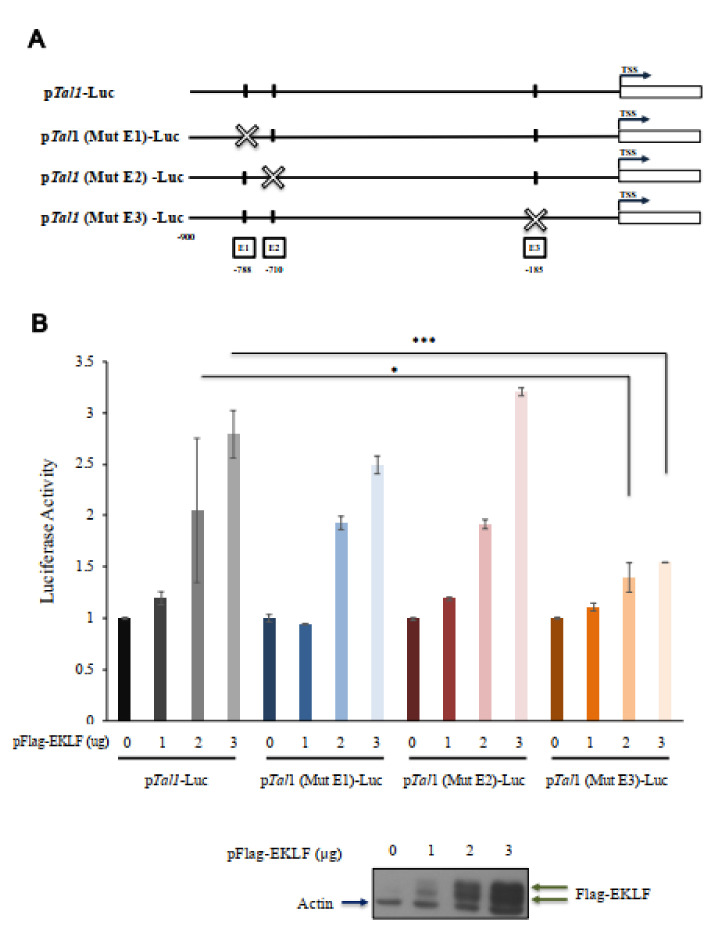
Transactivation of the *Tal1* promoter by EKLF. (**A**) Linear maps of the wild type and three mutant fragments, in which the CACCC box E1, E2, or E3 was mutated, and used for construction of the reporter plasmids p*Tal1*-Luc, p*Tal1*(Mut E1)-Luc, p*Tal1*(Mut E2)-Luc, and p*Tal1*(Mut E3)-Luc, respectively. (**B**) Luciferase reporter assay of the *Tal1* promoter in 293 cells. The dose-dependence of the luciferase (Luc) activity on the amount (ug) of pFlag-EKLF used in co-transfection is shown in the bar diagram. * *p* < 0.05, *** *p* < 0.001 by *t* test. Error bars: STD. The elevated levels of the exogenous Flag-EKLF upon co-transfection with increased amounts of the pFlag-EKLF plasmid were validated by immunoblotting.

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
