# Peer review of "A Positive Regulatory Feedback Loop between EKLF/KLF1 and TAL1/SCL Sustaining the Erythropoiesis"

_ijms, 2021, doi:10.3390/ijms22158024_

Round 1
Reviewer 1 Report
The erythroid Krüppel-like factor is a hematopoietic transcription factor that participates in the regulation of erythroid differentiation. Using a combination of microarray-based gene expression profiling and promoter-based ChIP-chip assay of fetal liver cells from wild type and Krüppel-like factor-knockout mouse embryos, together with molecular and cellular analyses of mouse erythroleukemic cells, the authors provide data establishing the existence of a previously unknown positive regulatory feedback loop between two DNA-binding hematopoietic transcription factors that sustains the mammalian erythropoiesis.
The experiments were carried out meticulously and the results were well presented. This paper provides important information regarding the production and differentiation of the erythroid lineage.
Author Response
The erythroid Krüppel-like factor is a hematopoietic transcription factor that participates in the regulation of erythroid differentiation. Using a combination of microarray-based gene expression profiling and promoter-based ChIP-chip assay of fetal liver cells from wild type and Krüppel-like factor-knockout mouse embryos, together with molecular and cellular analyses of mouse erythroleukemic cells, the authors provide data establishing the existence of a previously unknown positive regulatory feedback loop between two DNA-binding hematopoietic transcription factors that sustains the mammalian erythropoiesis. The experiments were carried out meticulously and the results were well presented. This paper provides important information regarding the production and differentiation of the erythroid lineage.
A: We thank this reviewer for his/ her encouraging comments.
Reviewer 2 Report
The authors described that among the downstream direct target genes of erythroid Krüppel-like factor (EKLF) is Tal1/Scl. EKLF activates Tall gene through binding mainly to the proximal promoter CACCC box E3. TAL1 is a known activator of Eklf gene transcription, while EKLF also positively regulates Tal1 gene transcription during erythroid differentiation from CFU-E/ pro-erythroblasts to the basophilic / polychromatic erythroid cells. Outcome of this study is the existence of a positive feed-back loop between the erythroid-enriched transcription factors, EKLF and TAL1, in early erythroid differentiation. The topic of the study is very interesting but major comments have to be addressed.
Major comments:
- The introduction is rather long with lots of details. The Introduction should be shorter and a more focused background to the current study, with the aim at the end of the introduction (not results) including short description in the proposed study.
- The discussion could more clearly highlight what is new in this study. It has rather long parts building up a picture based on various references. This is fine for a review, but in an original research article the own results should take more place. This is to give impression of confirming what other studies already have reported.
- The Results section looks like Discussion, to many references and descriptions. Please present clear and linked results of the study.
- In Results section, it is not clear do we have Tables or Supplemental Tables? And their order is questionable, we have first mentioned Table S2A (line 134), than Table S2B, C (line 142) and after Table 1 (line 173). However, we don't have mentioned Table S1A. Later on, we have mentioned Tables 2 (line 186), 3 (line 191) and 4 (line 207). Where are they? We don't have them in manuscript and supplements. We have Supplementary Table S3A and S3B in line 176, and other tables later on non-consistently presented. Very confusing.
Minor comments:
- Check consistency and order of references.
- The manuscript in its present form is rather difficult to follow for a reader, which unfortunately makes both the rationale and the novelty of the study unclear.
Author Response
The authors described that among the downstream direct target genes of erythroid Krüppel-like factor (EKLF) is Tal1/Scl. EKLF activates Tall gene through binding mainly to the proximal promoter CACCC box E3. TAL1 is a known activator of Eklf gene transcription, while EKLF also positively regulates Tal1 gene transcription during erythroid differentiation from CFU-E/ pro-erythroblasts to the basophilic / polychromatic erythroid cells. Outcome of this study is the existence of a positive feed-back loop between the erythroid-enriched transcription factors, EKLF and TAL1, in early erythroid differentiation. The topic of the study is very interesting but major comments have to be addressed.
Major comments:
- The introduction is rather long with lots of details. The Introduction should be shorter and a more focused background to the current study, with the aim at the end of the introduction (not results) including short description in the proposed study.
A: Indeed. We have since shortened the Introduction significantly, with the background literature introduction more focused.
- The discussion could more clearly highlight what is new in this study. It has rather long parts building up a picture based on various references. This is fine for a review, but in an original research article the own results should take more place. This is to give impression of confirming what other studies already have reported.
A: Totally agree. This is a very helpful suggestion. We have now deleted the details of the relevant literature at several place of Discussion and emphasized more the discussions of our data in relation to those literatures.
- The Results section looks like Discussion, to many references and descriptions. Please present clear and linked results of the study.
A: Agree. We have taken care of this as well.
- In Results section, it is not clear do we have Tables or Supplemental Tables? And their order is questionable, we have first mentioned Table S2A (line 134), than Table S2B, C (line 142) and after Table 1 (line 173). However, we don't have mentioned Table S1A. Later on, we have mentioned Tables 2 (line 186), 3 (line 191) and 4 (line 207). Where are they? We don't have them in manuscript and supplements. We have Supplementary Table S3A and S3B in line 176, and other tables later on non-consistently presented. Very confusing.
A: Terribly sorry about this ignorance during preparation of the manuscript. For all the supplementary tables, we now re-name them, according to the order of their appearance in the text, as Table S1A, Table S1B, Table S1C, Table S2, Table S3A, Table S3B, etc.
Minor comments:
- Check consistency and order of references.
A: Sorry. This has been taken care of now.
- The manuscript in its present form is rather difficult to follow for a reader, which unfortunately makes both the rationale and the novelty of the study unclear.
A: Agree. As outline above, we have since revised significantly the text to highlight the new findings and significance of this study.

Round 2
Reviewer 2 Report
Accept for publication